# Diversity-Aware Policy Optimization for Large Language Model Reasoning

**Jian Yao[1], Ran Cheng[1,2,3] ***, **Xingyu Wu[1], Jibin Wu[1,2], Kay Chen Tan[1]**

[1] Department of Data Science and Artificial Intelligence, The Hong Kong Polytechnic University
[2] Department of Computing, The Hong Kong Polytechnic University
[3] The Hong Kong Polytechnic University Shenzhen Research Institute, Shenzhen, China
nigel97.yao@connect.polyu.hk, {ran-peter.cheng, xingy.wu, jibin.wu, kctan}@polyu.edu.hk

## Abstract

The reasoning capabilities of large language models (LLMs) have advanced rapidly, particularly following the release of DeepSeek-R1, which has inspired a surge of research into data quality and reinforcement learning (RL) algorithms. Despite the pivotal role diversity plays in RL, its influence on LLM reasoning remains largely underexplored. To bridge this gap, this work presents a systematic investigation into the impact of diversity in RL-based training for LLM reasoning, and proposes a novel diversity-aware policy optimization method. Across evaluations on 12 LLMs, we observe a strong positive correlation between the solution diversity and Potential@k (a novel metric quantifying an LLM's reasoning potential) in high-performing models. This finding motivates our method to explicitly promote diversity during RL training. Specifically, we design a token-level diversity and reformulate it into a practical objective, then we selectively apply it to positive samples. Integrated into the R1-zero training framework, our method achieves a 3.5% average improvement across four mathematical reasoning benchmarks, while generating more diverse and robust solutions. The code is available at `https://github.com/nigelyaoj/R1_zero_Div`.

## 1 Introduction

Recently, the reasoning capabilities of large language models (LLMs) have made remarkable progress, with significant improvements showcased by OpenAI-o1 [35], DeepSeek-R1 [13], and Kimi-k1.5 [43]. Among these advancements, two key innovations have contributed significantly: First, the adoption of a rule-based reward system significantly streamlines the training process by focusing exclusively on rewarding correct final answers and proper output formats, thereby eliminating the complexity associated with process-based reward models [30, 47]. Second, the introduction of a lightweight reinforcement learning (RL) algorithm [13, 43] removes the need for a separate critic model, substantially reducing computational overhead and accelerating the training process. The success of DeepSeek-R1 has attracted numerous follow-up studies [62], which broadly fall into two categories. The first category focuses on improving the quality of training data [33, 15, 21, 2], emphasizing rigorous data set curation through filtering, deduplication, and verification. The second category refines RL algorithms, including detailed optimizations for PPO-based methods such as VCPPO [56] and VAPO [55]; enhancements to GRPO for stability and speed, such as DAPO [54], Dr.GRPO [31] and SRPO [63]; as well as alternative approaches such as REINFORCE++ [20].

While RL has been extensively applied to LLM reasoning, the role of diversity remains largely unexplored in this context, even though it plays a crucial role in RL research[19, 11, 36, 8, 37, 32, 64,

---

*Correspondence author

39th Conference on Neural Information Processing Systems (NeurIPS 2025).

12, 58, 66, 6, 52]. In traditional RL tasks, incorporating diversity is widely recognized to facilitate exploration by promoting the selection of more stochastic policies, which helps the policy escape local optima and accelerate the convergence of training. This hypothesis has been experimentally validated in previous work [19, 11, 36]. Beyond empirical evidence, theoretical analyses suggest that policies with higher entropy (a measure of diversity) can smooth the optimization landscape [1]. These findings naturally lead us to ask the following question: **Is promoting diversity essential during RL training for LLM reasoning?**

Intuitively, an LLM capable of generating diverse responses could broaden the exploration of reasoning paths, enabling the model to avoid overfitting to narrow solution patterns in mathematical or logical tasks. To formally address this question, we conduct an evaluation of diversity in LLM reasoning, with a specific focus on mathematical problem-solving. We introduce a novel metric, Potential@k, to quantify an LLM's reasoning potential (the possible performance gain after RL training). We empirically analyze 12 representative LLMs, examining both their solution diversity and Potential@k scores. Notably, our results reveal a strong positive correlation between solution diversity and Potential@k scores among high-performing models, which suggests that diversity directly contributes to improved final performance after RL training.

The empirical findings motivate us to promote diversity during RL training for LLM reasoning. A commonly used approach for this goal is entropy regularization. However, directly increasing the average entropy of LLM outputs can introduce length bias, as longer responses inherently exhibit higher entropy. To address this, we introduce a token-level diversity metric and reformulate the diversity objective into a practical form. Moreover, promoting diversity often entails a quality-diversity trade-off. To mitigate this, we strategically apply diversity enhancement only to positive samples, thereby enriching solution diversity while preserving training stability. This design is akin to fostering diversity in high-quality policies in population-based RL training, ensuring that exploration is guided by task-relevant performance criteria [48]. Finally, we integrate our diversity objective into the R1-zero training method and evaluate the enhanced approach across 4 mathematical reasoning benchmarks. Experimental results demonstrate a 3.5% average performance gain over standard R1-zero training, while our method can generate more diverse solutions.

To summarize, our key contributions are:

- We present the first formal investigation into the role of diversity in LLM reasoning. Through experiments on mathematical benchmarks, we identify a positive correlation between solution diversity and an LLM's reasoning potential, as measured by our proposed Potential@k metric. This finding provides empirical motivation for incorporating diversity into policy optimization.

- We propose a novel token-level diversity objective, which is reformulated into a practical metric and selectively applied to positive samples. This design is further supported through gradient behavior analysis, offering an insight for balancing quality and diversity during optimization.

- We evaluate our method on four mathematical reasoning benchmarks, each comprising at least 500 problems with stable evaluation metrics. Our method achieves a 3.5% average improvement over standard R1-zero training and consistently produces more diverse solutions.

## 2 Preliminary

### 2.1 RL for LLMs

In the context of RL for LLMs, we frame the LLM generation process as an RL problem. Here, the LLM is modeled as a policy that produces outputs (actions) conditioned on input prompts (states) and receives evaluative feedback (rewards) for its generated responses. This formulation aligns the sequential decision-making nature of language generation with RL's state-action-reward framework, enabling systematic optimization of the model's behavior through reward signals.

Formally, in the context of LLM generation for mathmatical problem-solving, where each prompt is a question, we define the prompt as $q \in \mathcal{Q}$, where $\mathcal{Q}$ represents the set of all possible questions. The set of all potential text outputs $o$ forms an action space $\mathcal{O}$. Each output $o$ consists of tokens, denoted

as $o := (o^1, o^2, ..., o^t, ...)$. To generate an output, a policy $\pi_\theta(\cdot|q)$ parameterized by $\theta$ is employed, which generates the output according to the distribution:

$$\pi_\theta(o|q) := \prod_t \pi_\theta(o^t|q, o^{<t}), \tag{1}$$

where $o^{<t} = (o^1, o^2, ...o^{t-1})$.

## 2.2 Reinforcement Learning algorithm

The R1-zero training method proposed by DeepSeek-R1 [13] has attracted significant research attention due to its computational efficiency and effectiveness. In our work, we adopt this training method as our backbone. R1-zero incorporates two key innovations: the GRPO algorithm [40] and a rule-based reward function. In this section, we introduce both components.

**Group Relative Policy Optimization (GRPO)**    GRPO streamlines the process by eliminating the need for a separate critic model, which is usually as large as the policy model, and instead estimates baselines using group scores. Specifically, for each question $q$, GRPO samples a group of outputs $\{o_1, o_2, ..., o_G\}$ from the old policy $\pi_{old}$ and optimizes the policy $\pi_\theta$ by maximizing the following objective:

$$J_{GRPO}(\pi_\theta) = \mathbb{E}_{q\sim\mathcal{Q},\{o_i\}_{i=1}^G\sim\pi_{old}(\cdot|q)}$$
$$\frac{1}{G}\sum_{i=1}^G \left( \min\left( \frac{\pi_\theta(o_i|q)}{\pi_{old}(o_i|q)}A_i, \text{clip}\big(\frac{\pi_\theta(o_i|q)}{\pi_{old}(o_i|q)}, 1-\epsilon, 1+\epsilon\big)A_i \right) - \beta\mathbb{D}_{KL}(\pi_\theta||\pi_{ref}) \right), \tag{2}$$

where $\epsilon$ and $\beta$ are hyperparameters, the KL term is defined as

$$\mathbb{D}_{KL}(\pi_\theta||\pi_{ref}) = \frac{\pi_{ref}(o_i|q)}{\pi_\theta(o_i|q)} - \log\frac{\pi_{ref}(o_i|q)}{\pi_\theta(o_i|q)} - 1, \tag{3}$$

and the advantage $A_i$ is computed using a group of rewards $\{r_1, r_2, ..., r_G\}$:

$$A_i = \frac{r_i - \text{mean}(\{r_1, r_2, ..., r_G\})}{\text{std}(\{r_1, r_2, ..., r_G\})}. \tag{4}$$

**Reward functions**    In line with DeepSeek-R1 [13], we implement two types of rule-based rewards: accuracy rewards and format rewards. The accuracy reward model assesses whether the response is correct by comparing the predicted answer to the golden reference answer, while the format reward model ensures that the final answer is presented in a \boxed{} format for reliable verification.

## 3   Correlation between LLMs' reasoning potential and solution diversity

The role of diversity has long been established as critical in traditional RL tasks. Numerous studies [19, 11, 36, 8, 37] have shown that promoting diversity can enhance the final quality of the policy. However, its impact in the realm of RL for LLM reasoning still remains under-explored. In this section, we investigate the relationship between solution diversity and the reasoning abilities of LLMs on mathematical benchmarks. We adopt the equation diversity in prior work [49] to quantify the variety of solutions generated for mathematical problem-solving. For reasoning ability, we introduce a novel metric to evaluate an LLM's training potential (related to the performance gain achieved after RL training).

**Experimental setup**    We evaluate 12 LLMs on the MATH benchmark [16]. For each question, we calculate: (1) **Pass@1 accuracy** using greedy decoding, and (2) **Diversity** with (3) **Potential@k**, both evaluated from 16 sampled responses (temperature=0.9).

For diversity, we adopt the metric (denoted as Div-Equ) from prior work [49], which measures the ratio of distinct equations among the responses:

$$\text{Div-Equ} := \frac{1}{N}\sum_{i=1}^N \frac{|\mathcal{U}_i|}{|\mathcal{A}_i|}, \tag{5}$$

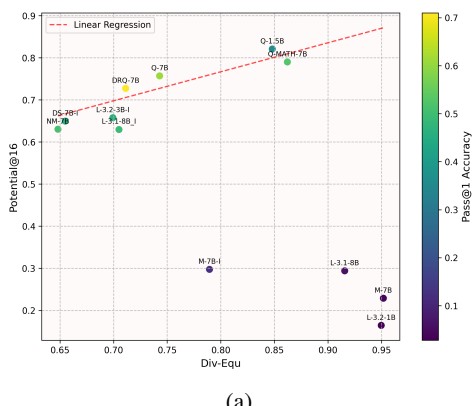
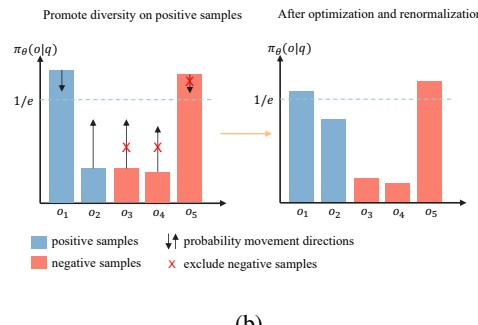

|(a)|(b)|

Figure 1: **(a)** Evaluation of Pass@1 accuracy, Div-Equ diversity, and Potential@16 across 12 LLMs on the MATH benchmark. Model naming conventions: Prefixes denote base architectures (Q: Qwen2.5-Math, DS: DeepSeekMath, M: Mistral, L: Llama, DRQ: DeepSeek-R1-Distill-Qwen, NM: NuminaMath); suffix '-I' indicates '-Instruct'. **(b)** Illustration of probability movement during diversity optimization on positive samples.

where $\mathcal{U}_i$ and $\mathcal{A}_i$ are the sets of unique equations and all equations extracted from the $k$ sampled responses (with $k = 16$ in our experiments) of question $i$, respectively. And $N = 500$ is the amount of the data.

For Potential, we define a metric termed Potential@k to quantify the model's capability to correct answers within $k$ trials (with $k = 16$ in our experiments) on its Pass@1 failure samples. Formally:

$$\text{Potential@k} := \frac{\sum_{i=1}^{N} \text{Pass@k}(q_i) \cdot (1 - \text{Pass@1}(q_i))}{\sum_{i=1}^{N} (1 - \text{Pass@1}(q_i))}, \tag{6}$$

where $q_i$ denotes the $i$-th question.

**Empirical findings** The results are shown in Figure 1a. The results show a bifurcated pattern: For LLMs with limited reasoning ability (Pass@1 < 0.4), we observe no significant relationship between solution diversity and model potential. For stronger performers (Pass@1 > 0.4), a clear positive correlation emerges between these metrics. Linear regression on this high-performing subset yields $R^2 = 0.81$, confirming a strong predictive relationship where increased diversity corresponds to higher model potential.

Through an investigation of the Objective 2 in the GRPO algorithm, we observe that for each question in the training set, if all samples within a group are either entirely positive or entirely negative, the advantage score becomes 0, resulting in no gradient update. Crucially, the training signal originates from the reward discrepancy between positive and negative samples within the group, which is inherently linked to our definition of potential (to some extent, the algorithm's improvement can be characterized by the dynamics of this potential metric, as discussed in Appendix B). This indicates that promoting diversity for LLM may result in higher performance after RL training.

**Takeaways** A positive correlation between the LLM's reasoning potential and solution diversity is observed in our experiment. As illustrated in Section 2.2, the optimization direction is guided by correct answers in multiple sampled responses. This directly links our Potential@k metric to RL training improvements. Hence, the observation strongly motivates us to enhance diversity during the RL training process.

## 4 Diversity-aware policy optimization

Building on the insights from Section 3, in this section, we introduce an entropy-based diversity and propose its targeted application to positive samples during policy optimization for LLM reasoning. We incorporate this diversity objective into the **R1-zero** training method [13], which employs the

GRPO algorithm with the reward function defined in Section 2.2. We refer to this enhanced approach as **R1-zero-Div**.

## 4.1 Entropy-based diversity

A straightforward approach is to define diversity as the average entropy of the LLM's outputs per question i.e., $E_{q \sim Q}[\mathcal{H}(\pi_\theta(\cdot|q))]$. However, this formulation introduces length bias: longer responses inherently exhibit higher entropy (due to more token-level uncertainties), causing the metric to artificially favor longer outputs regardless of actual solution diversity. To address this issue, we introduce token-level entropy, which calculates the entropy for each token sampled from the old policy $\pi_{\text{old}}$. Formally, we define:

$$\widehat{J}_{Div}(\pi_\theta) := \mathbb{E}_{q \sim \mathcal{Q}, o \sim \pi_{old}(\cdot|q)} \left[ \frac{1}{T} \sum_{t=1}^{T} \mathcal{H}(\pi_\theta(\cdot|q, o^{<t})) \right], \tag{7}$$

where $T$ is the length of the output.

During training, the gradient of diversity with respect to the policy $\pi_\theta$ in the $\mathcal{H}(\pi_\theta(\cdot|q, o^{<t}))$ is intractable. We therefore reformulate the diversity objective to enable effective backpropagation:

$$\widehat{J}_{Div}(\pi_\theta) = \mathbb{E}_{q \sim \mathcal{Q}, o \sim \pi_{old}(\cdot|q)} \left[ -\frac{1}{T} \sum_{t=1}^{T} \mathbb{E}_{\widetilde{o}^t \sim \pi_\theta(\cdot|q, o^{<t})} [\log \pi_\theta(\widetilde{o}^t|q, o^{<t})] \right]$$

$$= \mathbb{E}_{q \sim \mathcal{Q}, o \sim \pi_{old}(\cdot|q)} \left[ -\frac{1}{T} \sum_{t=1}^{T} \frac{\pi_\theta(o^t|q, o^{<t})}{\pi_{old}(o^t|q, o^{<t})} \log \pi_\theta(o^t|q, o^{<t}) \right]. \tag{8}$$

A proof for the last equation can be found in Appendix A.1. In practice, building on the R1-zero training method, we can use the samples within the group to calculate Objective 8.

## 4.2 Promoting diversity on positive samples

Empirical evidence indicates that the direct application of Objective 8 inadvertently increases diversity in incorrect solutions. Intuitively, negative samples offer more room for diversity enhancement, which can skew the model's optimization process. To address this issue, we concentrate on promoting diversity exclusively within positive samples:

$$J_{Div}(\pi_\theta) = \mathbb{E}_{q \sim \mathcal{Q}, o \sim \pi_{old}(\cdot|q)} \left[ -\mathbb{I}(r = 1) \cdot \frac{1}{T} \sum_{t=1}^{T} \frac{\pi_\theta(o^t|q, o^{<t})}{\pi_{old}(o^t|q, o^{<t})} \log \pi_\theta(o^t|q, o^{<t}) \right], \tag{9}$$

where $\mathbb{I}(\cdot)$ denotes the indicator function and $r$ is the accuracy reward for output $o$.

This is akin to fostering diversity in high-quality policies in population-based RL training [48], while we focus on positive samples rather than policies here. Beyond intuitive justification, we further justify this design by analyzing the gradient on each token.

According to Equation 8, we have:

$$\nabla_{\pi_\theta} \widehat{J}_{Div}(\pi_\theta) = \mathbb{E}_{q \sim \mathcal{Q}, o \sim \pi_{old}(\cdot|q)} \left[ -\frac{1}{T} \sum_{t=1}^{T} \frac{\nabla_\theta \left[ \pi_\theta(o^t|q, o^{<t}) \log \pi_\theta(o^t|q, o^{<t}) \right]}{\pi_{old}(o^t|q, o^{<t})} \right]. \tag{10}$$

Thus, the gradient can be decomposed into per-token contributions (each term in the summation contributes a component). Up to a constant scaling factor, the gradient from each token is:

$$-\nabla_\theta \pi_\theta(o^t|q, o^{<t}) \log \pi_\theta(o^t|q, o^{<t}) = -[1 + \log \pi_\theta(o^t|q, o^{<t})] \cdot \nabla_\theta \pi_\theta(o^t|q, o^{<t}). \tag{11}$$

Hence, for tokens with small probabilities (in that case $\pi_\theta(o^t|q, o^{<t}) < e^{-1}$, and this holds for most of tokens since the sum of probability is equal to 1), the gradient aligns with $\nabla_\theta \pi_\theta(o^t|q, o^{<t})$. This suggests that the diversity component's gradient actively promotes increasing the probability of

low-probability tokens, which inherently offer substantial growth potential. However, this tendency is undesirable for negative samples. Thus, excluding diversity enhancement for negative samples mitigates conflicts between solution quality and diversity. A visual illustration is provided in Figure 1b. Moreover, the experimental results in Section 5.3 and Appendix E.1 further support our design.

Finally, we incorporate the diversity optimization into the standard R1-zero training, and use the samples in the group to calculate the diversity, yielding the final training objective:

$$
\begin{aligned}
J(\pi_\theta) =& J_{GRPO}(\pi_\theta) + \lambda \cdot J_{Div}(\pi_\theta) \\
=& \mathbb{E}_{q \sim \mathcal{Q}, \{o_i\}_{i=1}^G \sim \pi_{old}(\cdot|q)} \frac{1}{G} \sum_{i=1}^{G} \Big[ \min \big( \frac{\pi_\theta(o_i|q)}{\pi_{old}(o_i|q)} A_i, \mathrm{clip} \big( \frac{\pi_\theta(o_i|q)}{\pi_{old}(o_i|q)}, 1 - \epsilon, 1 + \epsilon \big) A_i \big) \\
& - \beta \mathbb{D}_{KL}(\pi_\theta || \pi_{ref}) - \lambda \mathbb{I}(r_i = 1) \cdot \frac{1}{T_i} \sum_{t=1}^{T_i} \frac{\pi_\theta(o_i^t|q, o_i^{<t})}{\pi_{old}(o_i^t|q, o_i^{<t})} \log \pi_\theta(o_i^t|q, o_i^{<t}) \Big],
\end{aligned}
\tag{12}
$$

where $\lambda$ is the diversity weight and $i$ denotes the $i$-th sample in the group. In practice, we choose $\lambda = 0.01$. Other implementation details are provided in Section 5.1 and Appendix D.

## 5 Experiments

In this experimental section, we aim to address the following questions:

Q1. Can our method effectively enhance reasoning abilities and provide diverse solutions?

Q2. Does the design of the diversity coefficient $\lambda$ influence the results?

Q3. Does our method demonstrate consistent performance across different model sizes?

### 5.1 Experimental setup

**Base models**  We choose Qwen2.5-Math-7B (Qwen7B) [51] as our base model, which is commonly used for mathematical reasoning benchmarks [59, 67, 24]. Additionally, we conduct an ablation study using Qwen2.5-Math-1.5B (Qwen1.5B)[51] to assess the effectiveness of our approach in smaller LLMs.

**Benchmarks**  We selected 4 mathematical benchmarks to evaluate the models' reasoning abilities: GSM8K [7], MATH500 [16], Olympiad Bench [14], and College Math [42]. Each contains at least 500 data points for testing. We excluded some commonly used mathematical benchmarks that provide limited data, e,g, AIME24 [2] with 30 items, as they can lead to unstable and biased evaluation outcomes. We train the base model on the GSM8K training set and then evaluate on the 4 benchmarks.

**Baselines**  The most pertinent baselines for comparison are the base model itself and the base model trained via R1-zero. Additionally, we incorporate the latest prominent "R1-zero-Like" models with similar backbones **for reference**: SimpleRL-Zoo [59], PRIME-Zero-7B [9]. It is important to note that these methods are trained with different computational resources and datasets, making direct comparisons challenging. Our approach is designed to enhance diversity rather than compete directly with these methods. In fact, our method is compatible with and can be integrated into these existing approaches.

**Implementation details**  For R1-zero-Div, we train the base model on the GSM8K training set using the loss function in Equation 12, with a learning rate of $3 \times 10^{-6}$ and the AdamW optimizer. During rollout, we sample 6 responses with a temperature of 0.9 and train for 2 epochs. Our implementation is built on TRL [46] and runs on 8×A6000 GPUs. For R1-zero, we maintain identical settings to R1-zero-Div but exclude the diversity objective. For other baselines, we evaluate open-sourced models downloaded from Hugging Face[3], following the settings recommended in their original papers. Additional implementation details are provided in Appendix D.

---

[2]https://huggingface.co/datasets/Maxwell-Jia/AIME_2024
[3]https://huggingface.co

Table 1: Pass@1 accuracy on mathematical benchmarks.

| Method | GSM8K | MATH500 | Olympiad Bench | College Math | Avg |
|---|---|---|---|---|---|
| Qwen2.5-Math-7B | 57.6 | 51.8 | 16.1 | 21.4 | 36.7 |
| R1-zero | 88.7 | 74.6 | 37.3 | 43.3 | 61.0 |
| **R1-zero-Div (Ours)** | **91.7** | **78.2** | **40.1** | **47.6** | **64.4** |
| SimpleRL-Zoo | 90.2 | 80.0 | 39.0 | 47.2 | 64.1 |
| Eurus-2-7B-PRIME | 88.0 | 74.4 | 39.4 | 46.6 | 62.1 |

Table 2: Avg@8 accuracy on mathematical benchmarks.

| Method | GSM8K | MATH500 | Olympiad Bench | College Math | Avg |
|---|---|---|---|---|---|
| Qwen2.5-Math-7B | 53.37 (0.56) | 48.10 (0.82) | 15.80 (0.22) | 19.36 (0.14) | 34.16 |
| R1-zero | 87.77 (0.86) | 72.97 (1.20) | 37.26 (0.52) | 42.22 (0.31) | 60.06 |
| **R1-zero-Div (Ours)** | **90.64 (0.89)** | **76.92 (1.24)** | **39.19 (0.55)** | **47.49 (0.32)** | **63.56** |
| SimpleRL-Zoo | 89.46 (0.87) | 77.15 (1.23) | 39.43 (0.57) | 47.19 (0.34) | 63.31 |
| Eurus-2-7B-PRIME | 88.31 (0.86) | 73.92 (1.18) | 36.56 (0.50) | 45.27 (0.30) | 61.02 |

## 5.2 Main results

**R1-zero-Div enhances reasoning abilities**    We evaluate the reasoning performance using Pass@1 accuracy, as shown in Table 1. In our experiment, R1-zero-Div demonstrates superior performance compared to R1-zero, achieving an average improvement of $3.5\%$. Despite being trained with limited computational resources (discussed in Appendx B), R1-zero-Div achieves comparable results to state-of-the-art methods (SimpleRL-Zoo and Eurus-2-7B-PRIME). These results suggest that promoting diversity on positive samples in training can effectively enhance the model's reasoning capabilities. Also, following the recommendations in prior work [4, 17], we evaluated 8 samples per question with a temperature of $0.5$. We report Avg@8 and its standard error in the Table 2. The conclusion regarding the effectiveness of our approach remains consistent with the pass@1 metric results.

**R1-zero-Div generates diverse solutions**    We empirically demonstrate that R1-zero-Div produces more diverse solutions than other RL-finetuning baselines. Our evaluation on the GSM8K test set generates 5 responses for each of 1,319 questions, measuring diversity through three metrics: Div-Equ, and two additional metrics in prior work [26]: (1) N-gram diversity (proportion of distinct n-grams per response, capturing intra-diversity) and (2) Self-BLEU diversity (100 minus Self-BLEU score, capturing inter-diversity). All metrics range from 0 to 100, with higher values indicating greater diversity. As shown in Table 3, while RL fine-tuning methods significantly reduce diversity (compared to the base model), R1-zero-Div effectively preserves diversity. We further provide concrete examples in Appendix E showing that R1-zero-Div generates distinct solutions for the same question.

## 5.3 Ablation study

We conduct an ablation study to analyze (1) the impact of different diversity weights and (2) our method's generalization capability on smaller base models.

**Analysis on the choice of diversity weights** $\lambda$    Table 4 presents Pass@1 accuracy when applying different $\lambda$ values to promote diversity on positive samples (denoted as "pos"). The results demonstrate that small values ($\lambda \leq 0.02$) effectively enhance reasoning performance, with $\lambda = 0.01$ emerging as the optimal choice in our experimental setup. We further compare diversity promotion strategies: positive samples only ("pos") versus all samples ("pos+neg"). The marginal improvement observed when applying diversity to all samples supports our methodological design choice in Section 4.2.

Table 3: Diversity of different methods on GSM8K test set.

| Method | Div-Equ | Div-N-gram | Div-Self-BLEU |
|---|---|---|---|
| Qwen2.5-Math-7B | 92.26 | 29.29 | 85.98 |
| Eurus-2-7B-PRIME | 60.86 | 24.08 | 48.20 |
| SimpleRL-Zoo | 74.89 | 25.41 | 49.32 |
| R1-zero | 75.02 | 27.75 | 56.00 |
| **zero-Div (Ours)** | **79.29** | **29.60** | **58.89** |

**Experiment on 1.5B base model**   We perform both R1-zero-Div and R1-zero on the Qwen2.5-Math-1.5B base model [51], with results shown in Table 5. The experiments demonstrate that, compared to R1-zero, R1-zero-Div enhances reasoning performance on 3 out of 4 benchmarks, achieving an average improvement of 2.3%, validating the scalability of our approach to a smaller model.

Table 4: Ablation Study on different diversity weights on mathematical benchmarks

| Method | GSM8K | MATH500 | Olympiad Bench | College Math | Avg |
|---|---|---|---|---|---|
| $\lambda = 0$ | 88.7 | 74.6 | 37.3 | 43.3 | 61.0 |
| $\lambda = 0.05$, pos | 88.1 | 74.8 | 38.2 | 45.8 | 61.7 |
| $\lambda = 0.02$, pos | 90.7 | 76.0 | 38.4 | 45.9 | 62.8 |
| $\lambda = $ **0.01, pos** | **91.7** | **78.2** | **40.1** | **47.6** | **64.4** |
| $\lambda = 0.01$, pos+neg | 89.8 | 76.6 | 39.6 | 46.9 | 63.2 |

Table 5: Ablation Study on Qwen2.5-Math-1.5B base model

| Method | GSM8K | MATH500 | Olympiad Bench | College Math | Avg |
|---|---|---|---|---|---|
| Qwen2.5-Math-1.5B | 39.4 | 36.4 | 23.0 | 6.6 | 26.3 |
| R1-zero | 82.9 | 66.4 | **32.1** | 43.1 | 56.1 |
| **R1-zero-Div (Ours)** | **83.2** | **70.4** | 32.0 | **43.9** | **57.4** |

## 6   Related work

**RL for LLM reasoning**   The reasoning capabilities of LLMs have seen remarkable progress recently, with notable improvements demonstrated by OpenAI-o1 [35], DeepSeek-R1 [13], and Kimi-k1.5 [43]. Our work builds upon the R1-zero training method proposed by DeepSeek-R1 [13], which significantly improves LLM reasoning through two innovations that simplify the training pipeline and accelerate training: the GRPO algorithm [40], which replaces critic models with group score baselines, and a rule-based reward system that focuses solely on final answer correctness and output format.

Subsequent research has advanced this approach in two directions: (1) improving training data quality [28, 33, 15, 54, 21, 18] and (2) refining RL algorithms. Regarding RL algorithm refinement, one category focuses on PPO-like methods. SimpleRL-zero [60] demonstrates that PPO with replacing the reward model by a rule-based reward function can significantly improve the LLM's reasoning ability. VinePPO [25] leverages the flexibility of language environments to compute unbiased Monte Carlo-based estimates, eliminating the need for large value networks. VCPPO [56] employs a pretrained value model to address value initialization bias and decouples Generalized Advantage Estimation (GAE) computation between the actor and critic to mitigate reward signal decay. VAPO [55] further shows that value-based RL frameworks outperform value-free methods in long Chain-of-Thought

reasoning. The second category focuses on GRPO enhancements for stability and speed. DAPO [54] identifies the critical shortcomings (entropy collapse, training instability, and biased loss) in the original GRPO algorithm and addresses them via decoupled clipping and dynamic sampling. Dr.GRPO [31] reveals two biases in GRPO: response-level length bias and question-level difficulty bias. SRPO [63] introduces a two-stage history-resampling method to improve training efficiency. Alternative approaches explore algorithms like REINFORCE: Kimi-k1.5 [43] demonstrates stable training with REINFORCE-like policy gradients, while REINFORCE++ [20] and GPG [5] aim to enhance REINFORCE's stability and scalability, respectively.

**Diversity in RL**    Research on policy diversity in deep reinforcement learning can be categorized into three groups based on how diversity is utilized [48]. The first category uses diversity primarily to improve exploration efficiency, where diversity emerges as a byproduct of maximizing final task performance [19, 11, 36, 8, 37, 53]. The second category treats diversity either as a constraint (optimizing quality subject to diversity constraints) or as an objective (optimizing diversity under quality constraints) [32, 64, 12, 58, 66]. The third category optimizes quality and diversity simultaneously, known as Quality-Diversity RL methods [6, 38, 45, 3]. Our work extends the first paradigm to RL for LLM reasoning. While existing research in this category has proposed various diversity metrics, such as distance regularization between the current policy and a previous policy [19], reward randomization [41], we develop our approach based on a simple yet effective entropy-based diversity metric.

**Diversity in LLMs**    Prior work has explored diversity in LLMs across several domains. GEM [29] proposes methods to preserve diversity during supervised fine-tuning, while Bstar [61] investigates the exploration-exploitation tradeoff in self-improvement settings. Additional studies have examined diversity in reinforcement learning from human feedback [34, 26] and LLM ensembles [44]. However, diversity remains understudied in RL for LLM reasoning scenarios. To our knowledge, we are the first to formally analyze diversity and propose a principled diversity-aware training method for this setting.

## 7    Conclusion, limitations, and discussion

In this research, we investigate the role of diversity in RL for LLM reasoning. Through comprehensive evaluations across 12 LLMs, we empirically establish a strong positive correlation between a model's reasoning potential and the diversity of its generated solutions, underscoring the necessity of fostering diversity during RL training. To this end, we introduce a novel diversity-aware policy optimization method that optimizes the token-level diversity in positive samples. Experimentally, we demonstrate that our method not only enhances LLMs' reasoning ability but also generates more diverse solutions. By bridging the gap between diversity promotion and policy optimization, we aim to provide new insights for advancing the robustness and creativity of LLMs in complex reasoning scenarios.

Due to computational constraints, our experiments were conducted on $8\times$NVIDIA A6000 GPUs, which restricted our analysis to $1.5B$ and $7B$ parameter-scale models. This naturally introduces a limitation: the generalizability of our diversity-aware policy optimization method to larger-scale LLMs remains to be explored. While our method demonstrates significant improvements on mid-sized models, extrapolating these findings to larger architectures may require adjustments to the entropy regularization scheme or training dynamics, given the known differences in optimization landscapes across model scales. We urge future research to investigate these scalability challenges and hope our work will inspire the community to explore diversity-enhanced RL strategies for both small and large LLMs, fostering more robust reasoning capabilities across the spectrum of model architectures. Beyond scaling, the diversity-aware optimization mechanism could also be extended to other LLM tasks [65, 50, 23, 22], suggesting its broader applicability beyond reasoning tasks.

Another promising future direction lies in the semantic definition of diversity. In this work, we employ entropy-based regularization to implicitly promote diverse behaviors during LLM generation, which captures statistical variance in output distributions. However, many real-world applications demand user-intended diversity (e.g., requiring both algebraic and arithmetic solutions to a math problem, or generating code with distinct algorithmic approaches). Such scenario-specific diversity requires explicit modeling of user-defined diversity, a challenge well-studied in RL [10, 48]. By bridging LLM reasoning with explicit diversity optimization from RL, future work could unlock

more controllable and context-aware generative capabilities, addressing the gap between statistical diversity and human-intentional variety in complex tasks.

## Acknowledgments and Disclosure of Funding

This work was supported in part by the National Natural Science Foundation of China under Grant U21A20512 and the Research Grants Council of the Hong Kong SAR under Grant No. C5052-23G, Grant PolyU 15229824, Grant PolyU 15218622, and Grant PolyU 15215623. This work was also partially supported by Guangdong Basic and Applied Basic Research Foundation (No. 2024B1515020019). This work was also partially supported by the Research Grants Council of the Hong Kong SAR (Grant No. PolyU15217424, PolyU25216423) and the Hong Kong Polytechnic University (Project IDs: P0043563).

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

# A  Theoretical analysis

## A.1  Proof for Equation 8

The equation we want to prove is:

$$\mathbb{E}_{q\sim\mathcal{Q},o\sim\pi_{old}(\cdot|q)}\left[-\frac{1}{T}\sum_{t=1}^{T}\mathbb{E}_{\widetilde{o}^t\sim\pi_\theta(\cdot|q,o^{<t})}[\log\pi_\theta(\widetilde{o}^t|q,o^{<t})]\right]$$

$$=\mathbb{E}_{q\sim\mathcal{Q},o\sim\pi_{old}(\cdot|q)}\left[-\frac{1}{T}\sum_{t=1}^{T}\frac{\pi_\theta(o^t|q,o^{<t})}{\pi_{old}(o^t|q,o^{<t})}\log\pi_\theta(o^t|q,o^{<t})\right]. \tag{13}$$

Since $T$ is a random variable that depends on $\pi_{old}$, the proof is not straightforward. We prove it in two stages.

(1). When $T$ is fixed, the proof proceeds straightforwardly by examining each term in the summation. Note that $o^{<t}$ is sampled from $\pi_{\text{old}}$ while $\widetilde{o}^t$ is sampled from $\pi_\theta$, hence:

$$\mathbb{E}_{q\sim\mathcal{Q},o\sim\pi_{old}(\cdot|q)}\left[-\frac{1}{T}\sum_{t=1}^{T}\mathbb{E}_{\widetilde{o}^t\sim\pi_\theta(\cdot|q,o^{<t})}[\log\pi_\theta(\widetilde{o}^t|q,o^{<t})]\right]$$

$$=-\frac{1}{T}\sum_{t=1}^{T}\mathbb{E}_{q\sim\mathcal{Q},o\sim\pi_{old}(\cdot|q)}\left[\mathbb{E}_{\widetilde{o}^t\sim\pi_\theta(\cdot|q,o^{<t})}[\log\pi_\theta(\widetilde{o}^t|q,o^{<t})]\right]$$

$$=-\frac{1}{T}\sum_{t=1}^{T}\mathbb{E}_{q\sim\mathcal{Q},o^{<t}\sim\pi_{old}(\cdot|q)}\left[\mathbb{E}_{\widetilde{o}^t\sim\pi_\theta(\cdot|q,o^{<t})}[\log\pi_\theta(\widetilde{o}^t|q,o^{<t})]\right]$$

$$=-\frac{1}{T}\sum_{t=1}^{T}\mathbb{E}_{q\sim\mathcal{Q},o^{<t}\sim\pi_{old}(\cdot|q)}\left[\mathbb{E}_{o^t\sim\pi_{old}(\cdot|q,o^{<t})}[\frac{\pi_\theta(o^t|q,o^{<t})}{\pi_{old}(o^t|q,o^{<t})}\log\pi_\theta(o^t|q,o^{<t})]\right]$$

$$=-\frac{1}{T}\sum_{t=1}^{T}\mathbb{E}_{q\sim\mathcal{Q},o^{\le t}\sim\pi_{old}(\cdot|q)}\left[\frac{\pi_\theta(o^t|q,o^{<t})}{\pi_{old}(o^t|q,o^{<t})}\log\pi_\theta(o^t|q,o^{<t})\right]$$

$$=-\frac{1}{T}\sum_{t=1}^{T}\mathbb{E}_{q\sim\mathcal{Q},o\sim\pi_{old}(\cdot|q)}\left[\frac{\pi_\theta(o^t|q,o^{<t})}{\pi_{old}(o^t|q,o^{<t})}\log\pi_\theta(o^t|q,o^{<t})\right]$$

$$=\mathbb{E}_{q\sim\mathcal{Q},o\sim\pi_{old}(\cdot|q)}\left[-\frac{1}{T}\sum_{t=1}^{T}\frac{\pi_\theta(o^t|q,o^{<t})}{\pi_{old}(o^t|q,o^{<t})}\log\pi_\theta(o^t|q,o^{<t})\right]. \tag{14}$$

The second and fifth equations hold because we may add or remove any random variables that are not in the target expectation. The third equation results from applying importance sampling to reweight probabilities.

(2). For the case that $T$ is a random variable, roughly, the idea is to apply the law of total probability:

$$\mathbb{E}_{q\sim\mathcal{Q},o\sim\pi_{old}(\cdot|q)}\left[-\frac{1}{T}\sum_{t=1}^{T}\mathbb{E}_{\widetilde{o}^t\sim\pi_\theta(\cdot|q,o^{<t})}[\log\pi_\theta(\widetilde{o}^t|q,o^{<t})]\right]$$

$$=\mathbb{E}_{q\sim\mathcal{Q}}\left[\sum_{T_0=0}^{\infty}P(T=T_0)\mathbb{E}_{o\sim\pi_{old}(\cdot|q,T=T_0)}[-\frac{1}{T_0}\sum_{t=1}^{T_0}\mathbb{E}_{\widetilde{o}^t\sim\pi_\theta(\cdot|q,o^{<t})}[\log\pi_\theta(\widetilde{o}^t|q,o^{<t})]]\right] \tag{15}$$

And apply case (1) to finish the proof.

# B  More discussion

**More discussion about Potential@k**

The definition of Potential@k aims to quantify the performance improvement achievable through RL training for LLMs. By examining its formulation, we derive:

$$\text{Potential@k} := \frac{\sum_{i=1}^{N} \text{Pass@k}(q_i) \cdot (1 - \text{Pass@1}(q_i))}{\sum_{i=1}^{N}(1 - \text{Pass@1}(q_i))} \approx \sum_{i=1}^{N} [\text{Pass@k}(q_i) - \text{Pass@1}(q_i)]. \quad (16)$$

This metric essentially captures the discrepancy between Pass@k and Pass@1. While Pass@k is often treated as the performance boundary for RL training on LLM [57], our Potential@k specifically measures the performance gain from RL training, approximated by subtracting Pass@1 (a measure for initial performance) from Pass@k.

For each question $q_i$ before training begins, if $\text{Pass@1}(q_i) = 1$, the question is already mastered with no improvement potential. When $\text{Pass@1}(q_i) = 0$ but $\text{Pass@k}(q_i) = 1$, GRPO training uses positive samples from $k$ trials to teach the correct response. If both $\text{Pass@1}(q_i) = 0$ and $\text{Pass@k}(q_i) = 0$, the question provides no training signal as it remains unsolved. Hence, our definition of Potential@k focuses training on questions with partial capability, excluding both mastered and unsolvable questions, thereby capturing the true learning potential through the Pass@k to Pass@1 performance gap.

**Why the performance of our reproduction of R1-zero is worse than the state-of-the-art methods reproductions (e.g. SimpleRL-Zoo)?**

We believe the performance gap between our R1-zero reproduction and SimpleRL-Zoo's implementation stems primarily from resource constraints. Our experiments were conducted on a modest 8×A6000 GPUs setup, necessitating several efficiency optimizations: we employed the simpler GSM8K dataset, constrained generation lengths (appropriate for GSM8K's short responses). In contrast, SimpleRL-Zoo utilized significantly more powerful 2×8 H100-80G GPUs, trained on more complex datasets with longer response lengths. Importantly, our study's primary objective was not to surpass SimpleRL-Zoo's results, but rather to demonstrate that our diversity-enhanced method outperforms standard R1-zero. Our method of independence can be applied to enhance the SimpleRL-Zoo and other state-of-the-art methods.

**Why are the 4 mathematical datasets chosen?**

We require the number of data points in the test dataset to be at least 500. Since we find that the results are unstable when we test on a small dataset. In some recent work, they report by sampling many times and calculate the average. However, we think the distribution shift issue still exists (i.e., the small test data may biasly represent the hard/medium/easy-level benchmark)

## C    Broader impacts

Our diversity-aware RL approach for LLM reasoning offers valuable benefits for AI applications. First, in education, generating multiple valid reasoning paths could enhance AI tutoring systems by providing alternative solution strategies to students. Second, for scientific research, the improved ability to explore diverse reasoning approaches may aid in hypothesis generation and problem-solving where multiple perspectives are valuable.

## D    Implementation details

We provide more details for experiments in Section 5.

### D.1    Experiment environment

For training R1-zero and R1-zero-Div, the codebase runs on Python 3.11, utilizing TRL 0.16.0 [46] with PyTorch 2.5.1. We employ DeepSpeed [39] for distributed training and incorporate vLLM 0.7.2 [27] for efficient rollout, all deployed on 8× NVIDIA A6000 GPUs. Each experiment runs for 3 days. For other baselines, we evaluate open-sourced models downloaded from Hugging Face[4]

---

[4] `https://huggingface.co`

A conversation between User and Assistant. The user asks a question, and the Assistant solves it. The assistant first thinks about the reasoning process in the mind and then provides the user with the answer. The assistant should 1) Identify core concepts and required formulas. 2) Break down solutions into logical, numbered steps. 3) Verify results using alternative methods or substitutions. Put your final answer within \\boxed{}

Figure 2: System prompt

For evaluation, we utilize the code from Qwen2.5-Math. [5] To calculate Pass@1, we use greedy decoding for our models and baselines, except for SimpleRL-Zoo [59], which we evaluate using temperature=1 and top-p=0.95 as suggested in their paper.

## D.2 Hyperparameter settings

We provide the system prompt in Figure 2 and other detailed hyperparameter settings in Table 6. Due to computational resource constraints, we train on the simpler dataset (GSM8K), which allows for a shorter maximum response length, and use a well-designed prompt to obtain a stronger initial checkpoint. The experiment settings for R1-zero and R1-zero-Div are the same except for $\lambda = 0$ in R1-zero and $\lambda = 0.01$ in R1-zero-Div.

Table 6: Hyperparameter settings

| Hyperparameter | Value |
|---|---|
| *General settings* | |
| dataset | GSM8K |
| max prompt length | 256 |
| max completion length | 756 |
| num generations | 6 |
| use vllm | true |
| vllm gpu memory utilization | 0.5 |
| torch dtype | bfloat16 |
| learning rate | 3.0e-06 |
| lr scheduler type | cosine |
| beta | 0.0001 |
| zero stage | 2 |
| offload optimizer device | CPU |
| offload param device | none |
| distributed type | DEEPSPEED |
| *Base model: Qwen/Qwen2.5-Math-7B* | |
| num train epochs | 2 |
| per device train batch size | 1 |
| gradient accumulation steps | 64 |
| *Base model: Qwen/Qwen2.5-Math-1.5B* | |
| num train epochs | 3 |
| per device train batch size | 6 |
| gradient accumulation steps | 16 |

---

[5] https://github.com/QwenLM/Qwen2.5-Math

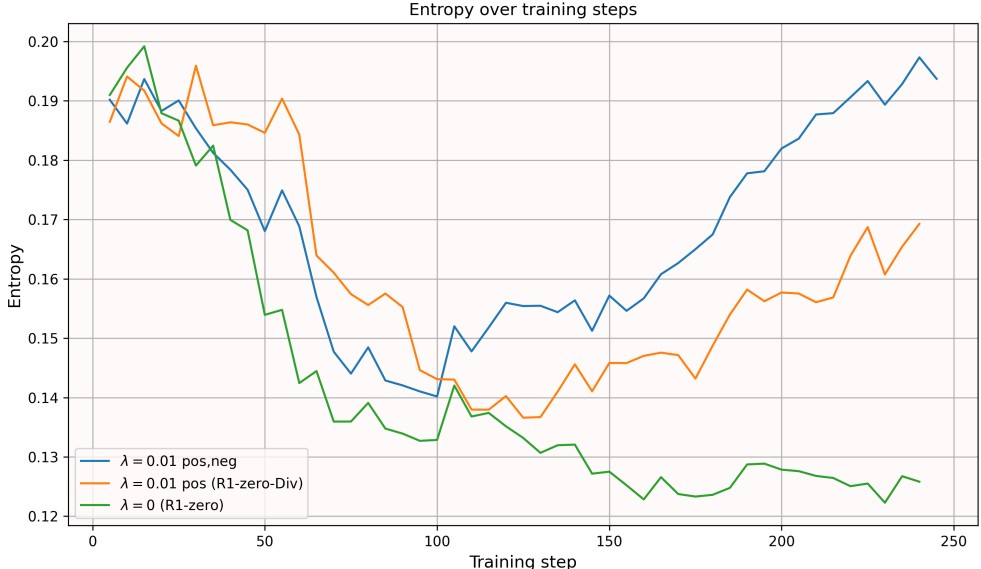

Figure 3: Entropy during the RL training

Table 7: Avg@8 on GPQA Diamond dataset

| Method | GPQA Diamond |
| --- | --- |
| Qwen2.5-Math-7B | 28.5 (0.64) |
| R1-zero | 32.7 (0.70) |
| **R1-zero-Div (Ours)** | **34.3 (0.79)** |
| SimpleRL-Zoo | 33.9 (0.56) |
| Eurus-2-7B-PRIME | 34.0 (0.74) |

## E   More experiment results

### E.1   Entropy during the RL training

We analyze the entropy dynamics during training. As shown in the Figure 3, when $\lambda = 0$ (i.e., the baseline R1-zero method), entropy collapses rapidly, indicating a loss of exploration. When applying the diversity objective with $\lambda = 0.01$ to all samples (both positive and negative), entropy exhibits late-stage exploration during training; however, this tends to degrade model quality, as reflected by the final performance in Table 4. In contrast, applying the diversity objective with $\lambda = 0.01$ exclusively to positive samples strikes a better balance between quality and diversity, yielding the optimal final performance.

### E.2   Evaluation on GPQA

We present the avg@8 results on the GPQA Diamond dataset in Table 7, where we directly evaluated the models listed in Table 1. These results demonstrate the generalizability of our method.

### E.3   Solution diversity in R1-zero-Div

We show the case where our method generates diverse solutions in Figure 4. The text outputs are generated by our model (R1-zero-Div training on Qwen2.5-Math-7B), and we render LaTeX expressions for better visualization. As illustrated in Figure 4, the model generates two distinct yet mathematically valid solutions to the problem. The first solution demonstrates elegant mathematical

Question: There exist real numbers $x$ and $y$, both greater than 1, such that $\log_x\left(y^x\right) = \log_y\left(x^{4y}\right) = 10$. Find $xy$.

Solution 1:

Given the equations:

$$\log_x(y^x) = 10 \quad \text{and} \quad \log_y(x^{4y}) = 10$$

1. First, we simplify the equations using logarithmic properties:

$$\log_x(y^x) = 10 \implies x \cdot \log_x(y) = 10 \implies \log_x(y) = \frac{10}{x},$$
$$\log_y(x^{4y}) = 10 \implies 4y \cdot \log_y(x) = 10 \implies \log_y(x) = \frac{10}{4y} = \frac{5}{2y}$$

2. Using the change of base formula, we know that $\log_x(y) = \frac{1}{\log_y(x)}$. Therefore:

$$\log_x(y) = \frac{10}{x} \quad \text{and} \quad \log_y(x) = \frac{5}{2y}$$
$$\frac{10}{x} = \frac{1}{\frac{5}{2y}} \implies \frac{10}{x} = \frac{2y}{5} \implies 10 \cdot 5 = x \cdot 2y \implies 50 = 2xy \implies xy = 25$$

Thus, the final answer is: $\boxed{25}$

Solution 2:

Given the equations:

$$\log_x(y^x) = 10 \quad \text{and} \quad \log_y(x^{4y}) = 10$$

we can use the properties of logarithms to simplify these equations.

1. Simplifying the first equation:

$$\log_x(y^x) = 10 \implies x\log_x(y) = 10 \implies \log_x(y) = \frac{10}{x}$$

Using the change of base formula $\log_x(y) = \frac{\log(y)}{\log(x)}$, we get: $\frac{\log(y)}{\log(x)} = \frac{10}{x}$

This implies: $\log(y) = \frac{10\log(x)}{x}$

2. Simplifying the second equation:

$$\log_y(x^{4y}) = 10 \implies 4y\log_y(x) = 10 \implies \log_y(x) = \frac{10}{4y} = \frac{5}{2y}$$

Using the change of base formula $\log_y(x) = \frac{\log(x)}{\log(y)}$, we get: $\frac{\log(x)}{\log(y)} = \frac{5}{2y}$

This implies: $\log(x) = \frac{5\log(y)}{2y}$

Now we have two equations:

$$\log(y) = \frac{10\log(x)}{x}, \log(x) = \frac{5\log(y)}{2y}$$

Substituting the second equation into the first:

$$\log(y) = \frac{10\left(\frac{5\log(y)}{2y}\right)}{x} = \frac{25\log(y)}{xy}$$

Dividing both sides by $\log(y)$ (assuming $\log(y) \neq 0$):

$$1 = \frac{25}{xy} \implies xy = 25$$

Thus, the value of $xy$ is: $\boxed{25}$

Figure 4: Solution generated by R1-zero-Div

insight by utilizing the reciprocal property of logarithmic functions. The second solution adopts a more systematic approach, first establishing two equations for $\log(y)$ and $\log(x)$ (more precisely, $\lg y$ and $\lg x$). And then solve these two equations. Both methods ultimately arrive at the correct solution, showcasing the model's ability to discover multiple valid reasoning paths.

