# OpenReview forum: "Diversity-Aware Policy Optimization for Large Language Model Reasoning"
_NeurIPS.cc/2025/Conference — NeurIPS 2025 spotlight_

### Official Review · Reviewer_kVwK · 2025-07-01

**Clarity:** 4
**Significance:** 3
**Originality:** 4
**Rating:** 5
**Confidence:** 5

**Summary:**

This paper demonstrate a empirical finding that response diversity is correlated with the RL training performance. The authors then design an auxiliary loss to optimize the diversity of responses, which enhance the performance of GRPO algorithm. The evaluation is solid and comprehensive, and the paper writing is clear and fluent. Good job!

**Questions:**

Major Concerns:
1. The empirical analysis demonstrate there is positive correlation between equation diversity and potential@k. However, when optimize the diversity of rollout responses, the author used entropy. Can you justify the gap between these two diversity metrics? What about assign higher weights to questions with high equation diversity during training? Or what's the relations between response entropy and potential@k?

2. Table 3, it seems that in proper lambda, the pos + neg is better than pos with lambda = 0.05 or 0.02. This indicates that the influence of lambda is more important than excluding negative samples. Could you provide the ablations between pos+neg and pos in all lambda settings?

Minor Concerns:
1. Please directly write out the model names in Table 1. The "base" and "base+" is not that straightforward.
2. Could you open-source your code?

**Ethical Concerns:**

["NO or VERY MINOR ethics concerns only"]

**Final Justification:**

5 accept. I have read all rebuttals and  the authors have answered my questions properly.

**Limitations:**

yes

**Quality:**

4

**Strengths And Weaknesses:**

Strengths:
The paper insightfully discovers the correlation between the diversity of responses under GRPO and the algorithm's effectiveness, and leverages this correlation to optimize the GRPO algorithm. It also achieves excellent results in the evaluation.

Weakness:
There are some minor flaws in writing, and the rest can be seen in the Questions.

---

> ### Author Rebuttal · Authors · 2025-07-31
>
> We are grateful to the reviewer for their perceptive comments and the time dedicated to assessing our work. We are greatly encouraged by the positive assessment (the paper is insightful, the evaluation is solid, and the writing is clear and fluent). To address the concerns raised, we conducted additional experiments and we provide detailed explanations point by point below.
>
> **Q1: The empirical analysis demonstrate there is positive correlation between equation diversity and potential@k. However, when optimize the diversity of rollout responses, the author used entropy. Can you justify the gap between these two diversity metrics? What about assign higher weights to questions with high equation diversity during training? Or what's the relations between response entropy and potential@k?**
>
> **A1**: Thank you for this insightful question. The two diversity metrics are strongly correlated：(1) We conduct a re-evaluation in empirical analysis using entropy as the diversity metric, and the result shows that potential@k exhibits a positive correlation with entropy (with $R^2 > 0.73$) in high-quality models, which is consistent with the correlation observed for equation diversity.
> (2) Optimizing entropy can also enhance equation diversity, as evidenced by results in Table 2 in the main paper. This confirms that entropy serves as a valid proxy for diversity in our setup. While equation diversity represents an **explicit** measure, direct optimization of such explicit metrics in RL is often challenging and unstable due to their discrete and task-specific nature. Hence, we choose entropy as a more general **implicit** metric, which aligns with diversity objectives and enables stable training.
>
> **Q2: Table 3, it seems that in proper lambda, the pos + neg is better than pos with lambda = 0.05 or 0.02. This indicates that the influence of lambda is more important than excluding negative samples. Could you provide the ablations between pos+neg and pos in all lambda settings?**
>
> **A2**: We provide the full results for $\lambda=0, 0.01, 0.02, 0.05$ under both pos and pos+neg settings in the table below. The results indicate that both the choice of $\lambda$ and the application of a diversity objective on positive samples are important. Specifically, the selection of $\lambda$ balances quality (the accuracy) and diversity, while applying a diversity objective on positive samples further stabilizes training and tends to converge to better quality.
>
> | $\lambda$ | GSM8K | MATH500 | Olympiad Bench | College Math | Avg |
> |---------------|-------|---------|-----------------|--------------|------|
> | 0 | 88.7 | 75.6 | 38.2 | 46.2 | 62.2 |
> | 0.01, pos | **91.7** | **78.2** | **40.1** | **47.6** | **64.4** |
> | 0.01, pos+neg | 89.8 | 76.6 | 39.6 | 46.9 | 63.2 |
> | 0.02, pos | 90.7 | 76.0 | 38.4 | 45.9 | 62.8 |
> | 0.02, pos+neg | 87.9 | 75.4 | 37.2 | 45.0 | 61.4 |
> | 0.05, pos | 88.1 | 74.8 | 38.2 | 45.8 | 61.7 |
> | 0.05, pos+neg | 86.7 | 75.8 | 36.7 | 44.5 | 60.9 |
>
>
> **Q3: Please directly write out the model names in Table 1. The "base" and "base+" is not that straightforward.**
>
> **A3**: Thank you for the suggestion. We will replace the abbreviations "base" and "base+" with the full model names in the revised version to ensure clarity.
>
> **Q4: Could you open-source your code?**
>
> **A4**: Yes, we will after the paper is accepted. We had uploaded the code for your review.
>
> We hope these revisions adequately address your concerns, and we are happy to provide any further clarifications to facilitate the review process!

---

> > ### Author Response · Authors · 2025-08-05
> > **A polite ask for reviewers' feedback on our rebuttal**
> >
> > Dear Reviewer kVwK,
> >
> > Thank you very much for your recognition of our work and for your valuable suggestions. We would like to kindly ask whether our responses have addressed your concerns. If there are any remaining questions or points that require further clarification, we would be more than happy to discuss them.
> >
> > Thank you once again for your valuable time and insightful feedback! We truly appreciate your thoughtful review and constructive suggestions, which have greatly helped us improve our work.

---

> ### Comment · Reviewer_kVwK · 2025-08-05
>
> Thanks for the authors' carefully responses. I have read the rebuttal and maintain my scores.

---

> > ### Author Response · Authors · 2025-08-06
> >
> > It is our pleasure, and many thanks for your insightful review! We will refine the paper according to your suggestions.

---

### Official Review · Reviewer_CURd · 2025-07-03

**Clarity:** 3
**Significance:** 2
**Originality:** 3
**Rating:** 4
**Confidence:** 3

**Summary:**

The paper investigates how encouraging solution diversity can boost RL of LLMs on reasoning tasks. Analyzing on mathematical benchmarks, the authors show a strong positive link between the potential@k metric and the diversity of generated answers, suggesting that more varied reasoning paths signal greater room to train. Building on this, they propose a token-level entropy regularizer that is applied only to reward-positive samples during GRPO training, so that it prevents length bias and limits quality-diversity trade-offs. Empirically the method lifts Pass@1 accuracy across math benchmarks with richer solution variety. Key contributions are the study of diversity in LLM reasoning and empirical evidence that diversity promotion reliably enhances several models’ reasoning performance.
.

**Questions:**

N/A

**Ethical Concerns:**

["NO or VERY MINOR ethics concerns only"]

**Final Justification:**

I have read all other reviews and authors' rebuttal. They have properly addressed my raised questions in weaknesses, and thus I've increased my rating from 3 to 4.

**Limitations:**

Yes, the authors discussed extensively in the conclusion section.

**Quality:**

3

**Strengths And Weaknesses:**

Strengths
- The authors provided clear empirical motivation to their method by showing that diverse reasoning paths signal larger RL head-room.
- Applying token-level entropy only to reward-positive samples is a reasonable approach.
- The empirical results demonstrated effectiveness of the proposed method on math benchmarks.

Weaknesses
- All experiments target math reasoning and Qwen2.5 models. It is unclear if the method/findings generalizes to other domains and models.
- In figure 1, potential@k correlates with diversity only when pass@1 > 0.4, and the metric itself is heuristic. There lacks a solid scientific explanation on the models with high diversity yet low performance and remains inconclusive to make a statement on diversity-performance relation.
- Relying on token-level entropy may still reward verbose yet redundant answers.
- (minor) Diversity-based exploration and token-level entropy are being heavily studied in recent literatures. The proposed method is reasonable but not entirely new.

---

> ### Author Rebuttal · Authors · 2025-07-31
>
> We thank the reviewer for their insightful comments and the valuable time dedicated to evaluating our work. We are encouraged by the reviewer’s recognition that our approach—applying token-level entropy exclusively to reward-positive samples—is reasonable and grounded in clear empirical motivation. To address the concerns raised, we have conducted additional experiments to further validate the generalizability of our approach, and we offer detailed point-by-point explanations below.
>
> **W1: All experiments target math reasoning and Qwen2.5 models. It is unclear if the method/findings generalizes to other domains and models.**
>
> **A1**:Thanks for this important feedback. We have conducted additional experiments across other reasoning domains and on other model:
>
> - (1) We evaluated our trained (on the math dataset) models on the GPQA Diamond benchmark [1], with results shown in Table A.
> - (2) We applied our approach to the Llama-3.2-3B-Instruct model on the K&K dataset [2], a logic reasoning task previously studied in LLM Reasoning research [3]. We trained on 3-person scenarios and evaluated on both 3-person (-3ppl) and 4-person (-4ppl) tasks (Table B).
>
> The results demonstrate that: (1) Improvements from enhanced diversity in mathematical domains generalize to graduate-level Q&A tasks (GPQA Diamond); (2) The approach is validated across both model architectures and reasoning domains.
>
> **Table A:  Evaluation on GPQA Diamond benchmark​ (Avg@8)**
>
> |               Method               | GPQA Diamond |
> |:----------------------------------:|:------------:|
> |           Qwen2.5-MATH-7B          |  28.5 (0.64) |
> |       Qwen2.5-Math-7B-R1-zero      |  32.7 (0.70) |
> | Qwen2.5-Math-7B-R1-zero-Div (Ours) |  **34.3 (0.79)** |
> | |
> |    Qwen2.5-Math-7B-SimpleRL-Zoo    |  33.9 (0.56) |
> |  Qwen2.5-Math-7B-Eurus-2-7B-PRIME |  34.0 (0.74) |
> ​
> **Table B:  Performance of Llama models on K&K benchmark​ (Avg@8)**
>
>
> |               Method              |   K&K-3ppl  |   K&K-4ppl  |
> |:---------------------------------:|:-----------:|:-----------:|
> |       Llama-3.2-3B-Instruct       |  8.8 (1.17) |  2.8 (0.39) |
> |   Llama-3.2-3B-Instruct-R1-zero   | 24.5 (1.47) | 17.3 (1.06) |
> | Llama-3.2-3B-Instruct-R1-zero-Div | **27.8 (0.95)** | **20.0 (0.75)** |
>
>
> **W2: In figure 1, potential@k correlates with diversity only when pass@1 > 0.4, and the metric itself is heuristic. There lacks a solid scientific explanation on the models with high diversity yet low performance and remains inconclusive to make a statement on diversity-performance relation.**
>
> **A2**: Thanks for the insightful feedback.
> - For "the metric itself is heuristic", we would like to kindly point out that, as we stated in Appendix B in the main paper, the metric potential@k essentially captures the discrepancy between Pass@k and Pass@1. While Pass@k is often treated as the performance boundary for RL training on LLM [4], our Potential@k specifically measures the performance gain from RL training, approximated by subtracting Pass@1 (a measure for initial performance) from Pass@k.
>  For each question $q_i$ before training begins, if Pass@1($q_i$) = 1, the question is already mastered with no improvement potential. When Pass@1($q_i$) = 0 but Pass@k($q_i$) = 1, GRPO training uses positive samples from k trials to teach the correct response. If both Pass@1($q_i$) = 0 and Pass@k($q_i$) = 0, the question provides no training signal as it remains unsolved. Hence, our definition of Potential@k focuses training on questions with partial capability, excluding both mastered and unsolvable questions, thereby capturing the true learning potential through the Pass@k to Pass@1 performance gap.
> - Explanation for models with high diversity but low performance: When quality is relatively low, exploration offers limited benefits to the model, this is supported by several studies in reinforcement learning (purely novelty-driven exploration struggles when the agent is essentially random [5]; when the policy is essentially random, the curiosity does not help learning of the extrinsic task[6]). Focusing on LLM reasoning, one extreme case is when a model’s samples contain no correct answers at all; no matter how high the diversity of its responses, it cannot access the correct answer.
>
> **W3: Relying on token-level entropy may still reward verbose yet redundant answers.**
>
> **A3**:
> - We deeply appreciate the reviewer’s astute observation regarding the potential for token-level entropy to incentivize verbose or redundant responses. However, this represents a challenging trade-off between quality and diversity that is inherent in such frameworks, as noted in prior RL literature [7,8,9]. Our approach mitigates this conflict by focusing token-level entropy exclusively on reward-positive samples, thereby aligning diversity with quality. This refinement is validated by our experimental results.
> - Moving forward, we recognize the value of exploring hybrid strategies that **further** balance diversity and quality, and one promising avenue could involve integrating efficient reasoning mechanisms [10,11] to optimize the generation of diverse yet concise responses.
>
>
> **W4: (minor) Diversity-based exploration and token-level entropy are being heavily studied in recent literatures. The proposed method is reasonable but not entirely new.**
>
> **A4**: Thanks for noting that. We would like to gently note that entropy-based methods have indeed gained attention in **very recent** studies. We will discuss these work in the final version.
>
> [1] GPQA: A Graduate-Level Google-Proof Q&A Benchmark
>
> [2] On Memorization of Large Language Models in Logical Reasoning
>
> [3] Logic-RL: Unleashing LLM Reasoning with Rule-Based Reinforcement Learning​
>
> [4] Does Reinforcement Learning Really Incentivize Reasoning Capacity in LLMs Beyond the Base Model?
>
> [5] Learning to Discover Skills through Guidance
>
> [6] Redeeming Intrinsic Rewards via Constrained Optimization
>
> [7] Curiosity-Driven Reinforcement Learning from Human Feedback
>
> [8] Exploring the Performance-Reproducibility Trade-off in Quality-Diversity
>
> [9] Quality-Diversity Actor-Critic: Learning High-Performing and Diverse Behaviors via Value and Successor Features Critics
>
> [10] Stop Overthinking: A Survey on Efficient Reasoning for Large Language Models.
>
> [11] Think When You Need: Self-Adaptive Chain-of-Thought Learning
>
>
> We hope these revisions adequately address your concerns, and we are happy to provide any further clarifications to facilitate the review process!

---

> > ### Comment · Reviewer_CURd · 2025-08-04
> >
> > Thank the authors for their rebuttal to address my raised questions. I will update my score accordingly.

---

> > > ### Author Response · Authors · 2025-08-05
> > >
> > > Thank you for the valuable suggestions provided during the rebuttal period, which have definitely improved the quality of our paper. We are happy to provide further explanation if any concerns remain.

---

### Official Review · Reviewer_FPC2 · 2025-07-03

**Clarity:** 3
**Significance:** 3
**Originality:** 3
**Rating:** 5
**Confidence:** 3

**Summary:**

This paper investigates the role of diversity in RL-based training for LLM reasoning and proposes a diversity-aware policy optimization method. The authors introduce Potential@k, a metric quantifying reasoning potential, and demonstrate a positive correlation between solution diversity and this metric across 12 LLMs. They propose a token-level diversity objective applied selectively to positive samples, integrated into R1-zero training. Experiments show 3.5% average improvement on mathematical reasoning benchmarks while generating more diverse solutions.

**Questions:**

- How does the method scale to larger models (>7B parameters) and longer reasoning sequences?
- Can the approach be extended to other reasoning domains beyond mathematics?
- What is the computational overhead of the diversity objective during training?

**Ethical Concerns:**

["NO or VERY MINOR ethics concerns only"]

**Final Justification:**

Rating 4.

the absolute gains appear modest;
the experiments are not verified on larger scale models

**Limitations:**

The authors acknowledge key limitations: computational constraints limiting model scale, focus on mathematical reasoning only, and unknown generalizability to larger architectures. The semantic definition of diversity (statistical vs. user-intended) is identified as future work.

**Quality:**

3

**Strengths And Weaknesses:**

* Strengths *
- Novel empirical insight: First systematic investigation of diversity's role in LLM reasoning with strong correlation evidence (R²=0.81)
- Well-motivated approach: Potential@k metric provides clear theoretical justification for diversity promotion
- Practical method: Token-level diversity objective avoids length bias and selective application to positive samples balances quality-diversity trade-off

* Weaknesses *
- Limited scale and scope: Only 1.5B-7B models tested; mathematical reasoning exclusively; computational constraints acknowledged
- Modest improvements: 3.5% average gain is meaningful but not dramatic; some individual benchmark improvements are marginal
- Theoretical gaps: Gradient analysis provides intuition but lacks formal guarantees; entropy regularization is standard approach

---

> ### Author Rebuttal · Authors · 2025-07-31
>
> We thank the reviewer for the perceptive comments and the time dedicated to evaluating our work. We are greatly encouraged by the positive assessment, particularly regarding the novel empirical insights, well-motivated approach, and practical utility of our work. To address the raised concerns, we have supplemented additional experiments and provide detailed responses point by point below.
>
> **W1: Limited scale and scope: Only 1.5B-7B models tested; mathematical reasoning exclusively; computational constraints acknowledged**
>
> **A1**: Thanks for the feedback. Regarding the model scale, while our current experiments focus on 1.5B-7B models due to computational constraints, we have extended our evaluations to include the Llama-3.2-3B-Instruct model (please refer to **Q2**), which further validates the approach's generalizability across model architectures. Scaling to larger models (>7B parameters) remains an important direction for future work, pending increased computational resources. As for the scope beyond mathematical reasoning, we address this in detail in our response to **Q2** below.
>
> **W2: Modest improvements: 3.5% average gain is meaningful but not dramatic; some individual benchmark improvements are marginal**
>
> **A2**: Following the Reviewer zoHB's suggestion, we have included the Avg@8 metric along with standard deviations in our results (Please refer to A2 in rebuttal to Reviewer zoHB). While the absolute gains appear modest, the improvements are statistically significant across key benchmarks. Notably, our method offers strong practical value: it requires few lines of code modifications to implement, making it highly efficient to integrate into existing RL pipelines.
>
> **W3: Theoretical gaps: Gradient analysis provides intuition but lacks formal guarantees; entropy regularization is standard approach**
>
> **A3**:
> - We sincerely appreciate the reviewer’s insightful comments on the theoretical aspects of our work. We acknowledge that formal theoretical guarantees are currently absent, particularly given the limited existing theoretical frameworks addressing the trade-off between diversity and quality optimization  (especially in the context of LLM). E.g.  "these models lack a natural means to control the inherent trade-off between quality and diversity" [1], "As such, the reward-diversity trade off typically relies on heuristics." [2], "without giving a mathematical theoretical analysis" [3].
> While such theoretical grounding remains sparse in the literature, we have endeavored to provide empirical rigor through comprehensive gradient sign analysis. Specifically, our analysis reveals that for negative samples, optimizing diversity gradients conflicts with quality objectives for the majority of tokens (at most 2 tokens are exceptions due to the summation of probability being 1).
> - While entropy regularization is a standard approach, our key contribution lies in identifying that vanilla entropy control exhibits a length bias (We have added an experiment in **A1** in the rebuttal to Reviewer zoHB). To address this, we introduce token-level entropy, reformulated into a tractable form, and strategically apply this modified entropy term exclusively to positive samples to enable more stable and effective training.
>
> **Q1: How does the method scale to larger models (>7B parameters) and longer reasoning sequences?**
>
> **A4**: Due to current computational constraints, our experiments have focused on models up to 7B parameters, as training and evaluating larger models (e.g., 13B or 70B) would require substantially more computational resources beyond our current capacity.​
> To validate the method’s generalizability across architectures, we extended our evaluations to include the Llama-3.2-3B-Instruct model (Please refer to Q2). Results from this additional experiment demonstrate consistent performance gains, confirming that our approach can effectively scale to alternative model architectures.​
> We recognize the value of testing larger models and longer sequences, which we plan to explore as future work once additional computational resources become available.​
>
> **Q2: Can the approach be extended to other reasoning domains beyond mathematics?**
>
> **A5**: Thanks for this important question. We have conducted additional experiments across other reasoning domains:
> - (1) We evaluated our trained (on math dataset) models on the GPQA Diamond benchmark [4], with results shown in Table A.
> - (2) We applied our approach to the Llama-3.2-3B-Instruct model on the K&K dataset [5], a logic reasoning task previously studied in LLM-RL research [6]. We trained on 3-person scenarios and evaluated on both 3-person (-3ppl) and 4-person (-4ppl) tasks (Table B).
>
> The results demonstrate that: (1) Improvements from enhanced diversity in mathematical domains generalize to graduate-level Q&A tasks (GPQA Diamond); (2) The approach is validated across both model architectures and reasoning domains.
>
> **Table A:  Evaluation on GPQA Diamond benchmark​ (Avg@8)**
>
> |               Method               | GPQA Diamond |
> |:----------------------------------:|:------------:|
> |           Qwen2.5-MATH-7B          |  28.5 (0.64) |
> |       Qwen2.5-Math-7B-R1-zero      |  32.7 (0.70) |
> | Qwen2.5-Math-7B-R1-zero-Div (Ours) |  **34.3 (0.79)** |
> | |
> |    Qwen2.5-Math-7B-SimpleRL-Zoo    |  33.9 (0.56) |
> |  Qwen2.5-Math-7B-Eurus-2-7B-PRIME |  34.0 (0.74) |
> ​
> **Table B:  Performance of Llama models on K&K benchmark​ (Avg@8)**
>
> |               Method              |   K&K-3ppl  |   K&K-4ppl  |
> |:---------------------------------:|:-----------:|:-----------:|
> |       Llama-3.2-3B-Instruct       |  8.8 (1.17) |  2.8 (0.39) |
> |   Llama-3.2-3B-Instruct-R1-zero   | 24.5 (1.47) | 17.3 (1.06) |
> | Llama-3.2-3B-Instruct-R1-zero-Div | **27.8 (0.95)**|  **20.0 (0.75)** |
>
>
>
> **Q3: What is the computational overhead of the diversity objective during training?**
>
> **A6**: As stated in Equation 12 in the main paper, our method leverages the sampled data from GRPO, eliminating additional sampling costs. The only overhead comes from gradient calculations for the diversity term, which in our environment adds no more than 2% to the training time compared to the R1-zero method.
>
>
> [1] Understanding the Quality-Diversity Trade-off in Diffusion Language Models
>
> [2] Effective Diversity in Population Based Reinforcement Learning
>
> [3] Quality-Diversity with Limited Resources
>
> [4] GPQA: A Graduate-Level Google-Proof Q&A Benchmark
>
> [5] On Memorization of Large Language Models in Logical Reasoning
>
> [6] Logic-RL: Unleashing LLM Reasoning with Rule-Based Reinforcement Learning​
>
>
> We hope these revisions adequately address your concerns, and we are happy to provide any further clarifications to facilitate the review process!

---

> > ### Comment · Reviewer_FPC2 · 2025-08-04
> >
> > Thank you for your replies. I have read them and checked the additional experimental results (Avg@8). I maintain my positive rating.

---

> > > ### Author Response · Authors · 2025-08-05
> > >
> > > Thank you for your recognition of our work and for the valuable suggestions provided during the rebuttal period, which have definitely improved the quality of our paper.

---

### Official Review · Reviewer_zoHB · 2025-07-10

**Clarity:** 3
**Significance:** 2
**Originality:** 2
**Rating:** 4
**Confidence:** 4

**Summary:**

The paper begins by presenting experiments that demonstrate a strong positive correlation between solution diversity and the potential performance improvements of LLMs trained using RL. Building on these findings, the authors propose a diversity-aware policy optimization approach, which augments the original RL training objective for correct responses with an entropy maximization term to encourage solution diversity. Experiments on Qwen2.5-Math-7B and Qwen2.5-Math-1.5B are conducted to validate the effectiveness of the proposed method.

**Questions:**

Please see the Weaknesses within Strengths and Weaknesses.

**Ethical Concerns:**

["NO or VERY MINOR ethics concerns only"]

**Final Justification:**

4: Borderline accept

**Limitations:**

yes

**Quality:**

2

**Strengths And Weaknesses:**

> **Strengths**

1. The paper grounds its proposed diversity-aware policy optimization approach in preliminary experimental results and mathematical analysis, supporting the idea that enhancing solution diversity can improve the performance of LLMs trained with RL. This makes the approach both promising and intuitively reasonable.

2. The paper evaluates the trained LLMs on four benchmarks and conducts ablation studies to validate the effectiveness of the proposed diversity-aware policy optimization approach.

> **Weaknesses**

1. Entropy control is a widely used technique in reinforcement learning, and the proposed diversity-aware policy optimization method introduces only minor modifications to the standard entropy maximization objective. However, the paper does not include a comparison with the baseline approach that uses vanilla entropy maximization.

2. The performance of the trained LLMs appears to be evaluated based on a single run, which may undermine the reliability of the experimental results [1]. To provide a more robust evaluation, especially when greedy decoding is not used, we recommend reporting the Avg@K metric, which averages performance scores over K independent runs.

**References**

[1] Incorrect Baseline Evaluations Call into Question Recent LLM-RL Claims

---

> ### Author Rebuttal · Authors · 2025-07-31
>
> We thank the reviewer for their insightful comments and valuable time spent on evaluating our work. We are encouraged by the reviewer’s recognition that our approach, grounded in preliminary experiments and mathematical analysis, is both promising and intuitively reasonable. To address the raised concerns, we have added additional experiments to further validate the robustness and effectiveness of our approach, and we provide detailed explanations point by point below.​
>
> **W1: Entropy control is a widely used technique in reinforcement learning, and the proposed diversity-aware policy optimization method introduces only minor modifications to the standard entropy maximization objective. However, the paper does not include a comparison with the baseline approach that uses vanilla entropy maximization.**
>
> **A1**:
> Thanks for your feedback. We would like to kindly point out that:
> - While entropy control is widely used in RL, its adaptation to RL for LLMs is non-trivial. Our key contributions to the algorithm are: We identify that vanilla entropy control exhibits a length bias. We address this by introducing token-level entropy, which we reformulate into a tractable form. This modified entropy term is then strategically applied to positive samples to further mitigate the conflict between quality and diversity and enable more stable and effective training.
> - We have conducted experiments using direct entropy regularization (i.e., encouraging $E_{q\sim Q} [H(\pi_{\theta}(·|q))]$) and report the results in the following table (labeled Qwen2.5-Math-7B-R1-zero-Div-vanilla). The result shows that vanilla entropy regularization yields only marginal improvements over Qwen2.5-Math-7B-R1-zero. Analysis of failure cases reveals that the model tends to generate longer, less meaningful responses, with higher output token entropy than other methods, which indicates its inability to balance entropy and response quality under vanilla entropy regularization.
>
> **W2: The performance of the trained LLMs appears to be evaluated based on a single run, which may undermine the reliability of the experimental results. To provide a more robust evaluation, especially when greedy decoding is not used, we recommend reporting the Avg@K metric, which averages performance scores over K independent runs.**
>
> **A2**: Thanks for the insightful suggestion to enhance evaluation stability. Following the recommendations in [1,2], we evaluated 8 samples per question with a temperature of 0.5. We report Avg@8 and its standard error in the table below. The conclusion regarding our approach’s effectiveness remains consistent with the pass@1 metric results. We acknowledge the potential for evaluation variance and note that our chosen benchmark includes over 500 test samples to further mitigate such variance.​ We will report both Avg@K and Pass@1 in the revision version.
>
> **Table: Avg@8 on mathematical Benchmarks.**
>
> |                Method               |      GSM8K      |     MATH500    |  Olympiad Bench | College Math |  Avg  |
> |:-----------------------------------:|:---------------:|:--------------:|:---------------:|:------------:|:-----:|
> |           Qwen2.5-Math-7B           |   53.37 (0.56)  |  48.10 (0.82)  |   15.80 (0.22)  | 19.36 (0.14) | 34.16 |
> |       Qwen2.5-Math-7B-R1-zero       |  87.77  (0.86)  | 72.97 (1.20)   |   37.26 (0.52)  | 42.22 (0.31) | 60.06 |
> | Qwen2.5-Math-7B-R1-zero-Div-vanilla |   88.58 (0.86)  |  73.32 (1.17)  |   35.91 (0.50)  | 44.79 (0.29) | 60.65 |
> |  Qwen2.5-Math-7B-R1-zero-Div (Ours) |   **90.64 (0.89)**  |  **76.92 (1.24)**  |   **39.19 (0.55)**  | **47.49 (0.32)** | **63.56** |
> | |
> |     Qwen2.5-Math-7B-SimpleRL-Zoo    |   89.46 (0.87)  |  77.15 (1.23)  |   39.43 (0.57)  | 47.19 (0.34) | 63.31 |
> |  Qwen2.5-Math-7B--Eurus-2-7B-PRIME  | 88.31 (0.86)    |  73.92 (1.18)  |   36.56 (0.50)  | 45.25 (0.30) | 61.01 |
>
> [1] Incorrect Baseline Evaluations Call into Question Recent LLM-RL Claims
>
> [2] A Sober Look at Progress in Language Model Reasoning:
> Pitfalls and Paths to Reproducibility
>
>
>
> We hope these revisions adequately address your concerns, and we are happy to provide any further clarifications to facilitate the review process!

---

> > ### Author Response · Authors · 2025-08-05
> > **A polite ask for reviewers' feedback on our rebuttal**
> >
> > Dear Reviewer zoHB,
> >
> > We would like to kindly ask whether our responses have addressed your concerns. If there are any remaining questions or points that require further clarification, we would be more than happy to discuss them.
> >
> > Thank you once again for your valuable time and insightful feedback! We truly appreciate your thoughtful review and constructive suggestions, which have greatly helped us improve our work.

---

> > ### Comment · Reviewer_zoHB · 2025-08-06
> >
> > Thank you to the authors for their detailed responses and the additional experimental results. My concerns have been addressed, and I will adjust my score accordingly.

---

> > > ### Author Response · Authors · 2025-08-06
> > >
> > > It is our pleasure, and many thanks for your insightful review! We will refine the paper according to your suggestions.

---

### Note · Authors · 2025-08-12

Dear AC and Reviewers,

Thank you for your time evaluating our paper and for your valuable feedback. We would like to summarize that: during the rebuttal, we

1. added results on other reasoning domains (GPQA, K&K) and additional models (LLaMA) to demonstrate the generalization of our approach
2. provided Avg@8 results in the main experiments for more robust evaluation (We will report Avg@8 in the revision version)
3. included further experiments (other $\lambda$s, trivial entropy baseline) and clarifications to better highlight the novelty and effectiveness of our method.

We sincerely thank you again for your constructive feedback and will revise the paper accordingly based on your suggestions.

Authors

---

### Decision · Program_Chairs · 2025-09-17

**Decision:**

Accept (spotlight)

**Comment:**

The paper establishes positive correlation between diversity in RL training and the LLM reasoning, and proposed a simple yet effective novel diversity-aware policy optimization method to encourage diversity among positive solutions. Experiments on Qwen2.5-Math-7B and Qwen2.5-Math-1.5B validates the effectiveness of the proposed method. Overall reviewers are happy with the motivation, writing and execution of the paper, with some minor ask to extend the experiments beyond math reasoning and qwen model family.